

# Unsupervised and supervised learning of interacting topological phases from single-particle correlation functions

Simone Tibaldi[1,2⋆], Giuseppe Magnifico[3,4,5], Davide Vodola[1,2†] and Elisa Ercolessi[1,2]

**1** Dipartimento di Fisica e Astronomia "Augusto Righi"
dell'Università di Bologna, I-40127 Bologna, Italy
**2** INFN, Sezione di Bologna, I-40127 Bologna, Italy
**3** Dipartimento di Fisica e Astronomia "G. Galilei",
Università di Padova, I-35131 Padova, Italy
**4** Padua Quantum Technologies Research Center, Università degli Studi di Padova
**5** Istituto Nazionale di Fisica Nucleare (INFN), Sezione di Padova, I-35131 Padova, Italy.

⋆ simone.tibaldi2@unibo.it , † davide.vodola@unibo.it

## Abstract

The recent advances in machine learning algorithms have boosted the application of these techniques to the field of condensed matter physics, in order e.g. to classify the phases of matter at equilibrium or to predict the real-time dynamics of a large class of physical models. Typically in these works, a machine learning algorithm is trained and tested on data coming from the same physical model. Here we demonstrate that unsupervised and supervised machine learning techniques are able to predict phases of a non-exactly solvable model when trained on data of a solvable model. In particular, we employ a training set made by single-particle correlation functions of a non-interacting quantum wire and by using principal component analysis, k-means clustering, t-distributed stochastic neighbor embedding and convolutional neural networks we reconstruct the phase diagram of an interacting superconductor. We show that both the principal component analysis and the convolutional neural networks trained on the data of the non-interacting model can identify the topological phases of the interacting model. Our findings indicate that non-trivial phases of matter emerging from the presence of interactions can be identified by means of unsupervised and supervised techniques applied to data of non-interacting systems.

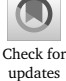

# 1   Introduction

In the last few years research in physics has seen a new series of methods and practices inspired by and exploiting machine learning [1]. These new instruments have proved to be useful in applications in many-body quantum physics [2], in particular for finding a representation of a wavefunction [3] and describing its dynamics [4], for reconstructing the wavefunction from experimental data [5], for speeding-up numerical simulations [4], and for classifying quantum phases of matter of both synthetic [6–11] and experimental data [12]. A particular type of the latter is given by symmetry protected and topologically ordered phases, whose classification escapes from the standard Landau theory of spontaneous symmetry breaking [13]. Indeed the combination of symmetries and topology can lead to new kinds of quantum phases that are characterized by a set of unusual features, such as *non-local* order parameters, the appearance of zero energy states at the boundary of the system, topological invariants, and long-range entanglement (for reviews see, for example: [14,15]). From an experimental point of view, topological materials have attracted much attention also because they represent a promising solution for physical implementation of qubits, more resilient to decoherence processes that affect devices based on superconducting or atomic physics technologies.

Due to their intrinsically non-local features and the lack of local order parameters, the classification of topological phases is considered a very challenging task to tackle [2]. However, machine learning methods have been successfully applied to both non-interacting models where the topological invariant (winding number) representing each phase was known a priori [16,17], and to interacting models where the topological invariant cannot be obtained easily [18].

Despite their success, in order to perform well, machine learning models require data sets with a very large number of training data, in the order of the thousands or millions. However, especially when handling interacting systems, it is not always easy to build such big data sets from both numerical simulations or experimental measurements. This led us to study a machine learning model trained on a data set obtained from a solvable system to be then applied to an interacting model which is obtained as an interacting generalization of the former, as in Ref. [19]. In this way, insights about the features of a simple dataset can be exploited to characterize the phases of a more complicated one, saving simulation resources. Differently from what has been done in previous works [16,20], our dataset will be constructed out of two-point single-particle correlation functions. These encode the properties of the different phases of the model and can be obtained numerically and measured experimentally [21–23], e.g. by using atom gas microscope with quenches to non-interacting models [24] or with randomized measurements [25].

In this work we show that supervised and unsupervised machine learning models can clas-

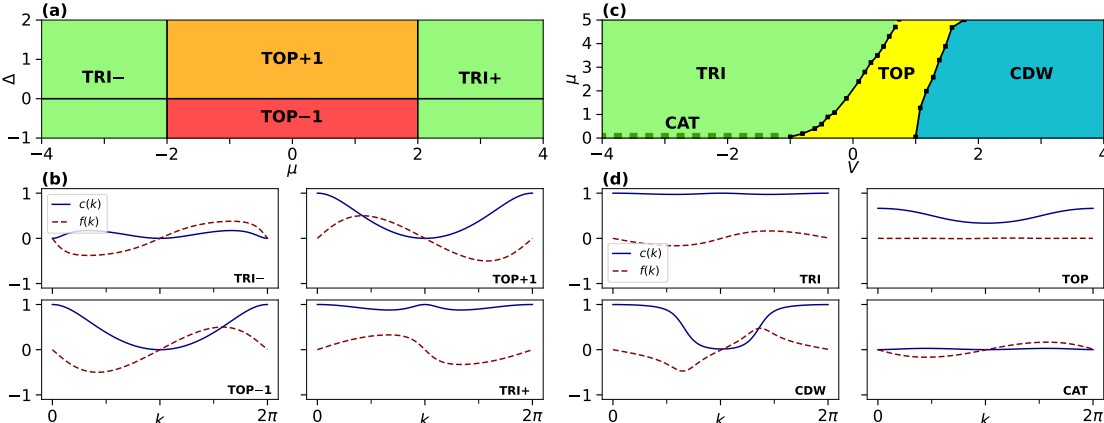

Figure 1: **Phase diagrams and correlation functions.** (a) Phase diagram of the non-interacting Kitaev model obtained from the winding number. The model presents three phases: A trivial phase with winding number $\nu = 0$ (green), that we distinguish for convenience in TRI- when $\mu < -2$ and TRI+ when $\mu > +2$, two superconducting topological phases with winding number $\nu = +1$ (TOP+1, orange) and $\nu = -1$ (TOP−1, red). The black lines represent phase transition lines. (b) Correlation functions of the non interacting model obtained from the analytical solution of the model. Each panel corresponds to a different region of the phase diagram. (c) Phase diagram of the interacting model. It presents four phases, of which three are trivial: Schrödinger-cat-like phase (CAT, dark green dashed line), charge-density wave (CDW, blue), trivial superconducting phase (TRI, green), and topological superconducting phase (TOP, yellow). The phase transition points (black squares) were obtained numerically by looking at the number of edge modes. (d) Correlation functions of the interacting Kitaev model obtained through the DMRG algorithm. The four graphs correspond to different regions of the phase diagram.

sify topological phases of an interacting system by being trained on data computed from simpler topological models. More specifically, our goal is to use machine learning techniques to predict the topological and non-topological quantum phases of two paradigmatic models: the one dimensional Kitaev chain in its non-interacting [26] and interacting scenario [27–32], with the idea that one can exploit the knowledge of the easily solvable non-interacting model in order to extract information also on the interacting case.

We construct the correlator dataset from the analytical solution of the non-interacting model and by means of numerical simulations implemented with the Density Matrix Renormalization Group (DMRG) algorithm for the interacting system [33]. Firstly, we probe the data with three unsupervised methods: the principal components analysis (PCA), k-means clustering and t-distributed stochastic neighbor embedding (t-SNE) [34–36]. With PCA we investigate to what extent the knowledge about principal component vectors of the data of the non-interacting case can be used to learn also the underline pattern that distinguishes the different phases in the interacting case. We use k-means' ability to find clusters and centroids to correctly predict the number of phases as well the location of the phase transition lines for both the non-interacting and interacting case. Then, we check that also t-SNE is able to form clusters of data.

Finally we devise an ensemble of convolutional neural networks (CNN) which is trained on the non-interacting data and then tested to predict the phases of the interacting model.

The paper is organized as follows: in Sec. 2, we introduce the models and the datasets that will be used for the training and testing of the machine learning methods. In Sec. 3,

we apply unsupervised methods (PCA, k-means clustering and t-SNE) to analyze the internal structure of the datasets. In Section 4, we apply a supervised model trained on the data of the non interacting model and test it on the interacting data. Finally, we draw our conclusions in Sec. 5.

## 2 Models

In this section we describe the two models, the non-interacting and the interacting Kitaev chain, whose quantum phases we want to classify, and we define the standard and anomalous correlation functions that will be used as indicator of the topological phases, thus providing the training and test sets.

### 2.1 Non-interacting topological superconductor

We consider the one-dimensional Kitaev model [26] defined on a lattice with $L$ sites described by the following non-interacting (NI) Hamiltonian

$$H^{\mathrm{NI}} = \sum_i (J a_i^\dagger a_{i+1} + \Delta\, a_i a_{i+1} + \mathrm{h.c.}) + \mu \sum_i a_i^\dagger a_i\,. \tag{1}$$

Here the operators $a_i$ ($a_i^\dagger$) annihilate (create) a spinless fermion on the lattice site $i$. The Hamiltonian $H^{\mathrm{NI}}$ describes a topological superconductor with nearest neighbour hopping of strength $J$, p-wave superconducting pairing of strength $\Delta$ and chemical potential $\mu$. When considering periodic boundary conditions, we can diagonalize $H^{\mathrm{NI}}$ by going to momentum space by means of standard Fourier transform, so that it is reduced to a sum over the Brillouin zone (BZ), $H^{\mathrm{NI}} = \sum_{k \in BZ} H(k)$, of a two-band Hamiltonian $H(k) = h_z(k)\sigma^z + h_y(k)\sigma^y$, where $k$ is the lattice quasi momentum and

$$h_z(k) = J\cos k + \mu/2\,, \quad h_y(k) = \Delta \sin k\,, \tag{2}$$

and $\sigma^x, \sigma^y$ are Pauli matrices. A Bogoliubov transformation casts $H^{\mathrm{NI}}$ in diagonal form $H^{\mathrm{NI}} = \sum_k E(k)\eta_k^\dagger \eta_k$, where $\eta_k$ are Bogoliubov operators and the single-particle energy $E(k)$ is given by

$$E(k) = \sqrt{h_z(k)^2 + h_y(k)^2}\,. \tag{3}$$

This model describes a one-dimensional topological superconductor belonging to the BDI symmetry class [37–40]. Its different phases are classified by the winding number $\nu$ of the normalized Hamiltonian vector $\hat{h}(k) = \vec{h}(k)/\|\vec{h}(k)\|$ with $\vec{h}(k) = (h_y(k), h_z(k))$, which is a continuous map from the 1D BZ to the circle $S^1$. The winding number $\nu$ is an integer that counts how many times the Hamiltonian vector $\hat{h}(k)$ turns around the origin when the quasi-momentum $k$ moves from 0 to $2\pi$ in the 1D BZ. Panel (a) of Fig. 1 shows the phase diagram of the model in Eq. (1) in the $(\mu, \Delta)$ plane, having set the energy scale $J = 1$. Notice that the phase diagram is symmetric for $\mu \leftrightarrow -\mu$. Three different phases appear: a trivial phase with winding number $\nu = 0$ (green) and two non trivial phases with winding number $\nu = \pm 1$ (orange/red). The winding number corresponds to the number of zero energy states that the model hosts at the boundaries of the chain, when considering open boundary conditions, a fact that is known in literature [14,41] as bulk-edge correspondence. For $\nu = 0$ no edge states will be present, while for $\nu = \pm 1$ an edge state on each boundary appears.

The information on the different phases can also be extracted from the Fourier transform

of the single-particle standard ($c(k)$) and anomalous ($f(k)$) correlation functions:

$$c(k) = \sum_{i,j} e^{ik(i-j)} \langle a_i^\dagger a_j \rangle \,, \tag{4}$$

$$f(k) = \sum_{i,j} e^{ik(i-j)} \langle a_i a_j \rangle \,, \tag{5}$$

where the expectation values are taken on the ground state. We note that $c(k)$ is real, while $f(k)$ is purely imaginary, due to the antisymmetry of the expectation value $\langle a_i a_j \rangle$ for the exchange $i \leftrightarrow j$. So we will redefine the latter by taking its imaginary part. For the Kitaev model of Eq. (1), $c(k)$ and $f(k)$ can be computed analytically and they take the form

$$c(k) = \frac{1}{2} + \frac{\mu/2 + J \cos k}{2E(k)} \,, \tag{6}$$

$$f(k) = \frac{\Delta \sin k}{2E(k)} \,. \tag{7}$$

Notice that they have a similar form to the components of the Hamiltonian vector $\hat{h}(k)$ from which the winding number is calculated. Their behaviour in the different phases is shown in panel (b) of Fig. 1.

The correlation functions $c(k)$ and $f(k)$ will be used for building a non-interacting training set $S$ where each data point is labelled with the winding number of its corresponding phase.

## 2.2 Interacting topological superconductor

We now add a nearest neighbor interaction term to the Hamiltonian (1) to obtain:

$$H^{\mathrm{I}} = \sum_i (J a_i^\dagger a_{i+1} + \Delta\, a_i a_{i+1} + \mathrm{h.c.}) + \mu \sum_i n_i + V \sum_i n_i n_{i+1} \,, \tag{8}$$

where $n_i = a_i^\dagger a_i$ is the occupation number at site $i$. This model cannot be solved exactly due to the interacting potential. By means of the DMRG algorithm [42], we have reproduced the phase diagram, after setting $J = \Delta = 1$, which is shown in panel (c) of Fig. 1, for $\mu > 0$ only since the model is symmetric for $\mu \leftrightarrow -\mu$. The model in Eq. (8) is characterized by only one topological superconducting phase (TOP, yellow) and three trivial phases: Topological trivial (TRI, green), Charge Density Wave (CDW, light-blue) and a Schrödinger-cat-like phase (CAT, light green dashed line) which shows up at the symmetric point $\mu = 0$ as a superposition of two trivial superconducting states with different occupation numbers [28–31]. At $V = 0$ we recover the non-interacting case with a critical point at $\mu = 2$. The different phases in Fig. 1(c) are detected from the number of edge states that appear in the chain: in the TRI, CDW and CAT phases the number of edge states is zero, while it is one in the TOP phase.

In the interacting case, it is not possible to evaluate exactly the correlation functions $c(k)$ (Eq. (4)) and $f(k)$ (Eq. (5)) on the ground state of the Hamiltonian of Eq. (8). Therefore we calculate them by means of the DMRG algorithm for a lattice of size $L = 100$. Some examples of the correlation functions for the different regions of the phase diagram are shown in panel (d) of Fig. 1.

# 3 Unsupervised training

Having obtained the datasets of the correlation functions for both the non-interacting and and interacting model, we use three unsupervised methods, namely PCA, k-means clustering and t-SNE, to extract the relevant information in both datasets and predict the phases of both models.

### 3.1 Principal Components Analysis

Principal components analysis is a standard technique used in statistics and machine learning for dimensionality reduction.

In order to apply PCA we start by creating a *design matrix*

$$X = \begin{pmatrix} c_1(k_0) & \dots & c_1(k_{L-1}) & f_1(k_0) & \dots & f_1(k_{L-1}) \\ & & \vdots & & & \\ c_N(k_0) & \dots & c_N(k_{L-1}) & f_N(k_0) & \dots & f_N(k_{L-1}) \end{pmatrix}, \tag{9}$$

where each of its $N$ rows is given by the correlation functions $c(k)$ and $f(k)$ from Eqs. (4) and (5) for one point $(\mu_\alpha, \Delta_\alpha)$, $\alpha = 1, \dots, N$, of the phase diagram. Each column represents the Fourier components of the correlation functions with quasi-momentum $k_n = 2\pi n/L$ ($n = 0, \dots, L - 1$) that are interpreted as the features of the data from which the PCA extracts the principal components. We rescale the columns of $X$ such that they have zero mean and unit standard deviation and we compute the eigenvalues $\{\lambda_i\}$, the explained variances $\epsilon_i = \lambda_i / \sum_j \lambda_j$, and the eigenvectors $\{\mathbf{w}_i\}$ of the correlation matrix $\mathcal{S} = X^T X$. The principal components of the data $X$, which are the eigenvectors of $\mathcal{S}$ corresponding to the largest eigenvalues, are the directions in a $2L$ dimensional space along which the original data show the largest variance. A common measure of the projection along the principal components is the *quantified leading component* (QLC) [6, 9]. For both the non-interacting and the interacting case, we compute the QLC by dividing the set of the two parameters in 40 sections, creating a grid of $40 \times 40$ subsets, each made of $M$ datapoints, as better specified below. Then, for each of the subsets we calculate the quantity

$$p_i = \sum_s \frac{|X_s \cdot \mathbf{w}_i|}{M}, \tag{10}$$

where $s$ runs on the elements of each subset, $X_s$ is the $s$-th row of the matrix $X$ corresponding to the $s$-th point of the subset, and $\mathbf{w}_i$ is the $i$-th eigenvector of $\mathcal{S}$. The value of the QLC for a datapoint shows how much that point is represented by that specific principal component.

*Non-interacting Hamiltonian* – For the Hamiltonian of Eq. (1), we create the desing matrix $X^{(\text{NI})}$ from data points generated in the range $\mu \in [-8, 8[$ and $\Delta \in [-2, 2[$ with a step of 0.1 for $\mu$ and 0.05 for $\Delta$, for a system with $L = 100$ sites. This subdivision of the ranges of $\Delta$ and $\mu$ corresponds to have a total of $N = 12800$ data points. The sum $\sum_{i=1}^4 \epsilon_i$ of the explained variances of the first four eigenvalues results in 96.6% meaning that most of the information of the data $X$ is contained in the first four eigenvectors. For this reason, in Fig. 2, we show the first four QLCs. These are calculated as in Eq. (10) by grouping the $N = 12800$ datapoints in $40 \times 40$ subsets corresponding to $M = 8$ datapoints per subset. The explained variances $\epsilon_i$ of each eigenvector are reported for each of the first four components. In panel (a) we see that $p_1$ is large for the points with $|\mu| > 2$, this means that the first principal component allows us to find the points of the phase diagram with winding number $\nu = 0$ (that belong to the trivial phases TRI− and TRI+). This is due to the shape of the first principal component $\mathbf{w}_1$ (depicted in the inset of Fig. 2(a)) that resembles the shape of the correlation functions $c(k)$ and $f(k)$ shown in Fig. 1(b) (TOP+1 phase). In Fig. 2(b), we see that, instead, $p_2$ allows us to extract the points of the phase diagram with winding number $\nu = \pm 1$ as the shape of the second principal component $\mathbf{w}_2$ (depicted in the inset of Fig. 2(b)) is similar to the correlation functions $c(k)$ and $f(k)$ shown in Fig. 1(b) (TOP±1 phases). This analysis shows that the first two principal components are sensitive to the trivial and non-trivial phases. All the other 198 QLCs have a total explained variance of the order of 13%, in particular the third QLC (Fig. 2(c)) has explained variance $\epsilon_3 = 5.2\%$ and recognizes the phase transition lines, while

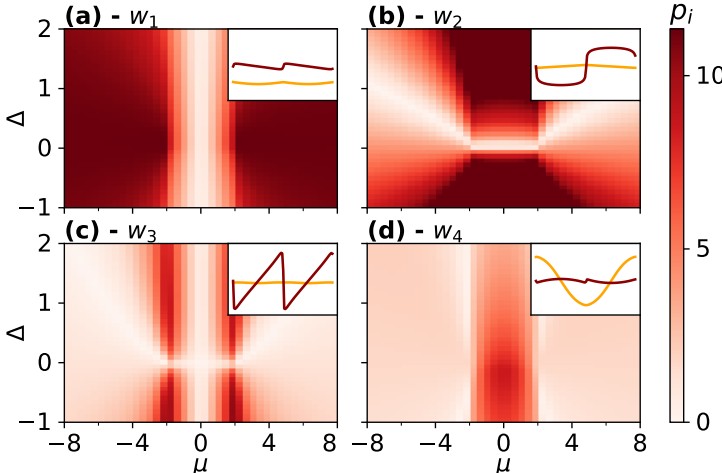

**Figure 2: PCA of the non-interacting Hamiltonian.** The quantified leading components (QLC) $p_i$ measure the projections of the data onto the space of each of the first four principal components, corresponding to the eigenvalues with largest explained variance $\epsilon_i$. (a) The first QLC with $\epsilon_1 = 0.451$ reveals the points of the phase diagram with trivial winding number. (b) The second QLC, with $\epsilon_2 = 0.421$, highlights the points of the phase diagram with non-trivial winding number, (c) The third QLC, with $\epsilon_3 = 0.052$ shows the phase transition lines, (d) The fourth QLC with $\epsilon_4 = 0.042$ has a high projection on the phase with $\nu = -1$. The insets show the first (red circles) and the second (orange triangles) 100 elements of the corresponding principal components $\mathbf{w}_i$ ($i = 1, 2, 3, 4$).

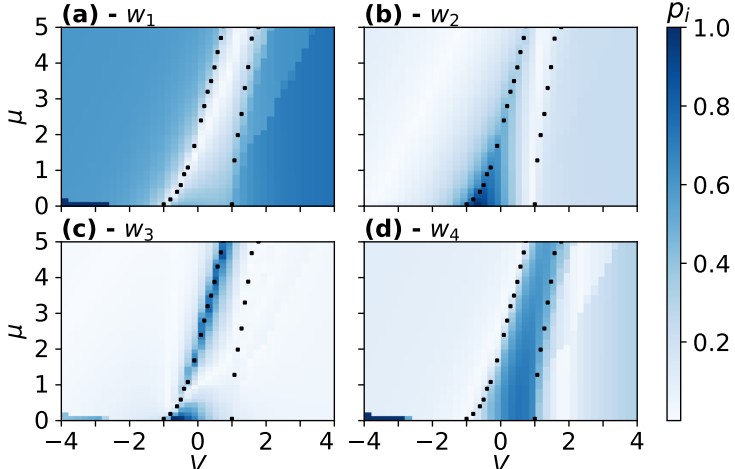

**Figure 3: PCA of the interacting Hamiltonian** Projection of the interacting correlation functions along the principal components obtained applying PCA to the non interacting dataset. Darker colors correspond to areas of the phase diagram which are more similar to the principal component $\mathbf{w}_i$ ($i = 1, 2, 3, 4$) used for the projection. QLCs drawn in (a) and (d) highlight the trivial (superconduting and charge density wave) and topological phases respectively. In both cases, the CAT phase is highlighted by higher values of $p_i$ compared to the other phases. Component (c) pins down a phase transition line whereas component (b) does not seem to be informative on the interacting dataset. The projections of all four plots are rescaled independently. The dots are the same as panel (c) of Fig. 1, added to help recognize the phase transition points.

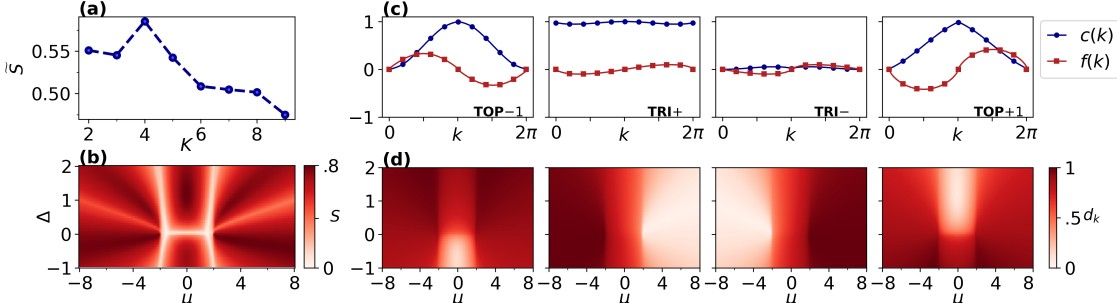

Figure 4: **Non-interacting Hamiltonian: k-means clustering.** (a) Values of the silhouette score $\widetilde{S}$ as a function of the number of clusters $K$. The maximum value is reached at $K = 4$ and suggests the best number of centroids for the dataset points. (b) Reconstructed phase diagram of the non-interacting model (with $K = 4$ centroids) through the average silhouette $S$ of every data point. The critical lines are clearly visible because the correlation functions of critical points have similar distances between two centroids. (c) Centroids obtained by applying k-means algorithm with $K = 4$. Every centroid resembles the correlation functions of the four different regions: topological with $\nu = -1$, the two trivial phases with $\nu = 0$ and topological with $\nu = +1$, from left to right. (d) Distances $d_k$ of the correlation functions for each point of the phase diagram from the corresponding $k$-th centroid (plotted in the corresponding panel above). The points belonging to the same phase show minimal distance to the same centroid.

the fourth (Fig. 2(d)) has explained variance $\epsilon_4 = 4.2\%$ and has a larger projection on the phase with winding $\nu = -1$.

*Interacting Hamiltonian* – We are interested in understanding if the principal components of the non-interacting model computed before can be used to distinguish among the phases of the interacting Hamiltonian. To this end, we construct a design matrix $X^{(I)}$ with data obtained from the interacting Hamiltonian in the range $\mu \in [0, 5[$, $V \in [-4, 4[$ with a step of 0.1. Then, we calculate the QLC by projecting these data along the principal components $\mathbf{w}_i$ of the non-interacting Hamiltonian. The first four QLC are shown in Fig. 3. The first (panel (a)) and the fourth (panel (d)) principal components $\mathbf{w}_1$ and $\mathbf{w}_4$ of the non-interacting data show higher projections on the interacting data and are thus able to discriminate between the trivial and topological phases, respectively. We will comment in Sec. 4 about the appearance of a ligther colored region in the CDW phase, just on the right of the phase transition line for large values of $\mu$. The third principal components $\mathbf{w}_3$ (panel (c)) recovers only the transition line between the trivial and the non-trivial superconducting phases while the second one seems to highlight only the region for low $\mu$ and $V \in [-1, 0]$. We note that, even though the CAT phase is present only on the single line $\mu = 0$, three out of four principal components are able to recognize it (Fig. 3(a), (c), (d)).

We can say with fairness that the PCA is able to learn the underlying pattern which distinguishes a trivial phase from a topological one. This is a good indication that even a supervised method can exploit the representation it learns of the non-interacting data to classify the interacting ones.

## 3.2 K-means clustering

K-means clustering is a machine learning method for finding clusters and cluster centers in a set of unlabelled data [35]. The algorithm starts by choosing a number of cluster centers called *centroids* and then it iteratively moves the centers to minimize the total variance within

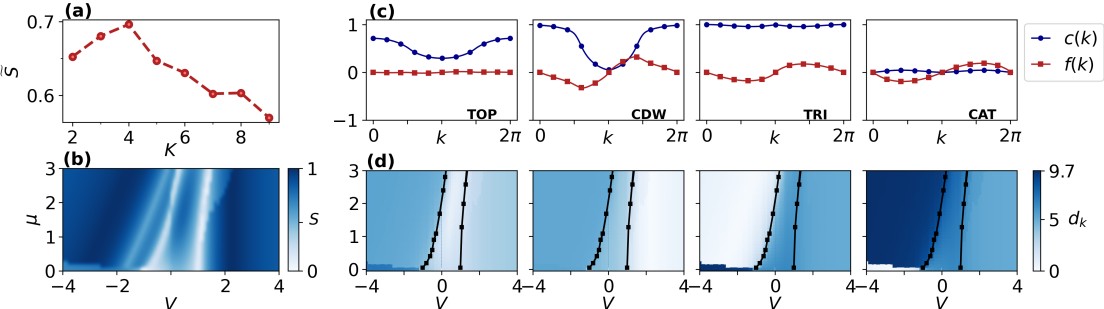

Figure 5: **Interacting Hamiltonian: k-means clustering** (a) Values of the silhouette score $\widetilde{S}$ as a function of the number of clusters $K$. The peak is at $K = 4$ in accordance to the 4 phases of the original phase diagram. (b) Reconstructed phase diagram of the interacting model (with $K = 4$ centroids) through the average silhouette value of every data point. Critical lines separating the phases are clearly visible, CAT phase is also highlighted. (c) Centroids obtained by applying k-means algorithm with $K = 4$. Every centroid resembles the correlation functions of the four different phases, as indicated by the label in the bottom right corner. (d) Distances of the correlation functions for each point of the phase diagram from the corresponding centroid (plotted in the panel above). The points belonging to the same phase show minimal distance to the same centroid.

cluster. The centroids are the central point of every cluster that are calculated by the algorithm autonomously, so we interpret them as the most representative point of the cluster.

In order to settle on a value of $K$ without any *a priori* knowledge of our data we choose to calculate the *silhouette score* $\widetilde{S}$ which is a value representing the average quality of the clusterization for each $K$. Calling $a$ the distance of a sample point to its centroid and $b$ its distance to the second closest centroid we can calculate the silhouette of a data point as $S = (b - a)/\max(a, b)$. The values for $S$ lie in the range $[-1, 1]$ where negative numbers correspond to a wrongly assigned point. Positive values indicate the right classification and the quality increases reaching 1.

We run k-means algorithm with different $K$ values and select the $K$ with the largest silhouette score. Due to the sensitivity of the algorithm to initial conditions we run k-means 10 times for each $K$ and then collect the average silhouette value $\widetilde{S}$ of every point over the 10 runs, whereas the centroids and projections correspond to the largest silhouette only.

*Non-interacting Hamiltonian* – The analysis of the silhouette for the non-interacting data turns out to be very informative. Firstly, after multiple iterations as explained above, we find the maximum value of the silhouette score at $K = 4$ as shown in Fig. 4(a). This suggests that the most reasonable way to divide the data points is in 4 classes, which might correspond to 4 different phases. Secondly, we see that the silhouette of a point can be used for identifying the phase transition lines. In fact, the points lying near a phase transition might be reasonably associable to the two different phases of the transition, that is they might show properties of both phases. So we expect their silhouette value to be close to 0, indicating that their classification might be non ideal. This is indeed the case, as it is seen from Fig. 4(b) which shows the average silhouette of every point calculated over 10 iterations of k-means algorithm applied with $K = 4$, as suggested by the previous analysis. We can see that points that are inside the phases have a larger silhouette than points lying at the phase transition.

Let us now move to the analysis of the centroids obtained by applying k-means algorithm with $K = 4$. In Fig. 4(c) and (d) we plot the 4 centroids and the distance of the points to each centroid, respectively. Each centroid seems to exactly represent the features of the datapoints,

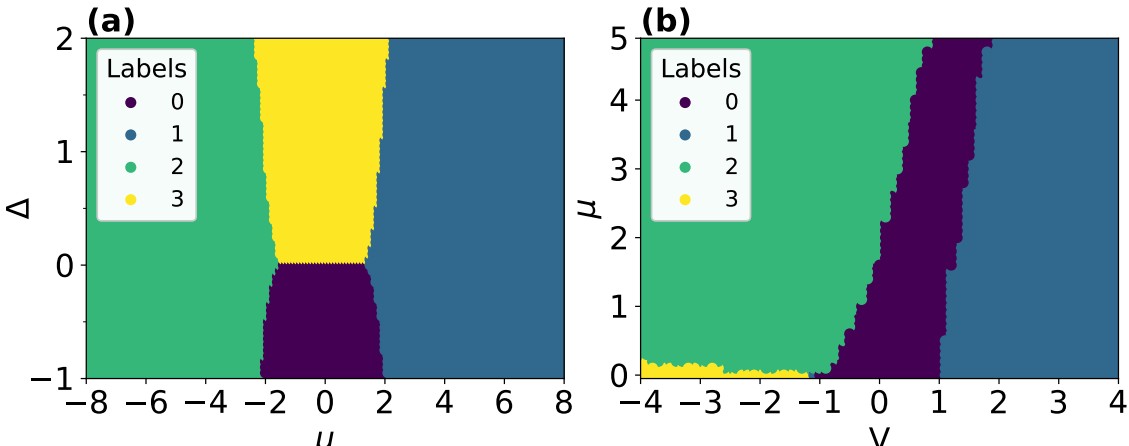

Figure 6: **Phase diagrams with k-means**. Reconstruction of the phase diagrams of the non-interacting (a) and the interacting (b) models obtained with the labels assigned by k-means. For each data point we plot its label predicted by k-means running with $k = 4$, the value chosen according to the best silhouette as explained in the text.

i.e. the correlation functions, of the four different regions of the phase diagram. Also, all the points of a phase have a distance very close to zero to their centroid, showing that we can separate the phases of the diagram with ease.

*Interacting Hamiltonian* – The same analysis can be repeated on the interacting dataset, obtaining similar results which are collected in Fig. 5. In particular the silhouette reaches its peak at $K = 4$ corresponding to the four phases of the interacting model shown in Fig. 1(c).

In Fig. 6, for completeness, we show the scatter plots of the labels assigned by k-means ran with $k = 4$ for both datasets. The reconstruction of the phase diagrams is neat and in accordance with the results of Figs. 4 and 5.

We can summarize the results of this section saying that, in both the non-interacting and the interacting models, the k-means clustering algorithm, by trying to separate points, is able to learn the characteristic shape of the correlation functions of each phase and to identify the boundaries where the phase transitions occur.

## 3.3 t-distributed stochastic neighbor embedding

t-SNE [36] is a popular dimensionality reduction technique that creates a low-dimensional distribution of data which is faithful to the original one and thus helps visualizing clusters of data in 2 or 3 space. It starts by creating a probability distribution from the Euclidean distances between data points in their original space. In particular given two points $x_i, x_j \in \mathcal{D}$ where $\mathcal{D}$ is the dataset, we interpret the conditional probability $p_{i|j}$ as a measure of similarity between the two points under a Gaussian centered at $x_i$:

$$p_{j|i} = \frac{\exp(||x_i - x_j||^2/2\sigma^2)}{\sum_{k \neq i} \exp(||x_i - x_k||^2/2\sigma^2)}, \tag{11}$$

where $\sigma$ is a parameter that we will discuss later. t-SNE creates a symmetric joint probability $P$ from 11 by taking $p_{ij} = (p_{i|j} + p_{j|i})/2$. Then, it gives to each point $x_i$ a new set of coordinates in a lower-dimensional space $y_i \in \mathbb{R}^d$ for $d = 2$ or 3, where the similarity between points is

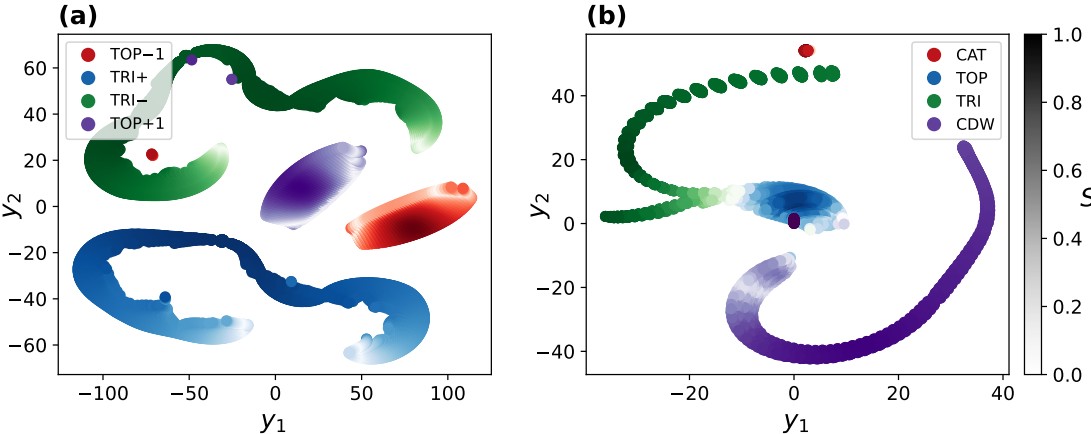

Figure 7: **Clustering with t-SNE**. The plot shows the visualization of the data projected in 2D applying t-SNE (components $y_1, y_2$). In both panels we assign a color to each point corresponding to its phase in the model while the brightness of each point corresponds to its silhouette (see Section 3.B). (a) Projection of the non-interacting model data points. Four clusters appear evidently and they correspond to the 3 phases of the non-interacting model with the two trivial phases (TRI+, TRI-) separated. The silhouette of the points (brightness of the color) highlights the points close to a phase transition, which happen to be at the borders of each cluster. (b) Projection of the interacting model data points. The separation of the clusters is less neat compared to (a) but thanks to the silhouette we can see the borders of each cluster.

given by a $t$-Student distribution $Q$ in the form:

$$q_{ij} = \frac{||y_i - y_j||^2}{\sum_{k \neq l} ||y_k - y_l||^2} .$$

(12)

In order to make $q_{ij}$ a faithful low-dimensional representation of the distribution $p_{ij}$, t-SNE minimizes the Kullback-Leibler divergence

$$C = KL(P||Q) = \sum_i \sum_j p_{ij} \log \frac{p_{ij}}{q_{ij}} .$$

(13)

Typically this is done updating the position of points $y_i$ by gradient descent:

$$y_i = y_i - \eta \frac{\partial C}{\partial y_i} ,$$

(14)

with $\eta$ the learning rate. The parameter $\sigma$ appearing in 11 is related to another hyperparameter called perplexity which is the one that needs to be set when running the t-SNE. Perplexity can be seen as the average number of neighbors we expect the points to have in the original space and typical values range from 5 to 50 [36].

We ran t-SNE on our datasets varying both perplexity and learning rate $\eta$ without seeing major changes to the final result. Also, no relevant distinction was found in either reducing the dimensions to 3 or 2, so we stick to 2D.

In Figure 7 we plot the results of t-SNE on the non-interacting/interacting dataset. For both models we plot the dimensional reduction to 2 dimensions. In the non-interacting case we see four well-separated clusters and each one corresponds to one of the phases of the model. In

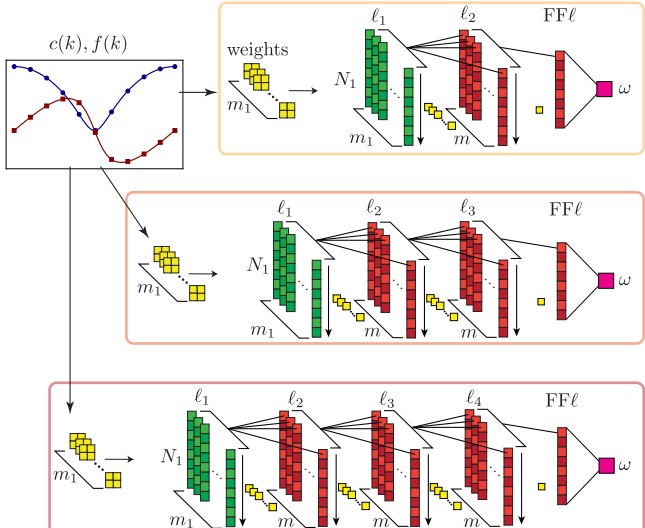

Figure 8: **Structure of the CNNs forming the ensemble**. The input for the CNNs is a $2 \times L$ dataset with the correlation functions $c(k), f(k)$. First, the input is filtered by a layer of $m_1$ 2D convolutional weights (in yellow) trained independently on each network. Each weight produces one array of length $N_1 = L - 1$ (the first set of green columns denoted by $\ell_1$) for a total of $m_1$ arrays. Then each network has a different number of additional hidden layers (columns in red): $\ell_2, \ell_3, \ell_4$ from the top. Each of these layers performs a 1D convolution using the weights in yellow. The last convolution always has one single weight in order to produce a single array. The last layer is a feed-forward layer (FF$\ell$) with one final output neuron that gives a real number which is the predicted topological indicator $\omega$ by the network. All the resulting topological indicators from the different CNNs that constitute the ensemble are then averaged to give the predicted topological indicator $\widetilde{\omega}$ (Eq. (15)).

particular t-SNE separates the two trivial phases (TRI+, TRI-) of the non interacting model in the same way as K-means does. For the interacting case the clusters are not as well-separated except for the CAT phase. Although the shape of the of the low-dimensional representation is not meaningful in t-SNE cluster, we are interested in finding the position of the points close to the phase transition lines. For this reason we adjust the brightness of each data point according to its silhouette value that was calculated in section 3.B (compare with Figs. 4 and 5). In this way we are able to see that the points closer to a phase transition are close to the margin of the cluster.

# 4 Supervised Training

In this section we exploit the information gained from the unsupervised analysis of the data to use a supervised model for predicting the different (trivial or topological) phases via the analysis of correlation functions. Our aim is to train the network on the data of the non-interacting Hamiltonian and test it on the interacting data. This is an approach that has been recently exploited in the context of machine learning applied to systems without analytical solution, e.g. [43].

For this reasons we decided to employ an ensemble method which is favourable even when tackling difficult classification problems [2]. The base element of the ensemble we use is a convolutional neural network (CNN), a type of machine learning model that has found many

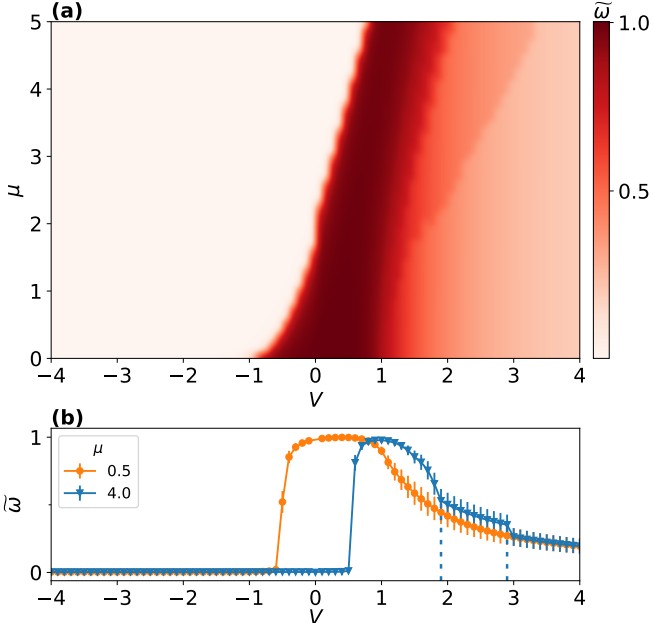

Figure 9: **Convolutional neural networks.** (a) Predictions $\widetilde{\omega}$ of the phase diagram of the interacting Hamiltonian carried out by means of the CNN ensemble. The topological phase is correctly predicted (dark red region). (b) Two cuts of the phase diagram showing the predictions $\widetilde{\omega}$ and their relative standard errors as a function of $V$ for different $\mu$. For the points with $\mu \gtrsim 1$ in the CDW phase it is possible to see the appearance of discontinuities (indicated by dashed vertical lines) for the values $\widetilde{\omega}$ that correspond to the onset of the incommensurate CDW phase.

successful applications in image recognition problems [44] and vastly employed in applications to quantum many body problems [16,17,45,46]. This network uses small matrices of weights, called features, which perform a series of discrete convolutions and are able to extract local properties of the data, for details see [34].

*CNN ensemble* – The ensemble of classifiers is made of $C = 180$ CNNs, as schematically shown in Fig. 8. Each CNN has an initial hidden layer $\ell_1$ (in green) of $m_1 = 100$ filters (weights) of size $2 \times 2$ (yellow in the picture) and stride 1, which produce $m_1 = 100$ arrays of size $N_1 = L-1 = 99$. Each network can have three different internal structures whose schemes are reported in Fig. 8. The number of additional hidden layers $\ell_\alpha$ (in red) can vary from 1 to 3 while each of the hidden layers in a network has the same number of weights per layer which can be $m = 25, 50, 100$. Finally, the networks have been trained with different batch sizes (512, 1024) and different random initial training configuration of the weights centered at zero (10 different starting seeds for each network). Each network accepts as input the $2 \times L$ matrix made by stacking $c(k)$ and $f(k)$ from Eqs. (6)-(7) and produces a single real number $\omega_\alpha$ in output which is the estimate for the topological indicator of the input. All layers neurons have ReLU activation ($f(x) = \max\{0, x\}$) except for the last one which is linear ($f(x) = x$). The loss function is given by the mean squared error. The output of each network is then averaged to produce:

$$\widetilde{\omega} = \frac{1}{C} \sum_{\alpha=1}^{C} \omega_\alpha. \tag{15}$$

In the following we will show that this quantity can be used as a topological indicator in order to reconstruct the phase diagram of the interacting model.

*Training* – The networks are trained separately. The dataset of the non-interacting model is

made of $2 \times 10^5$ points covering the whole phase diagram of the non interacting Hamiltonian of Fig. 1(a). Each network is trained with Adam gradient descent method by using an early stopping technique until convergence is achieved. This typically requires around 100 steps.

*Testing* – For the test dataset we considered $4 \times 10^3$ evenly spaced samples in the grid $[-4, 4] \times [0, 5]$ for $V$ and $\mu$ in order to fully reconstruct the interacting phase diagram of Fig. 1(c). The ensemble, trained on the non-interacting data, efficiently evaluates the topological classification of the test set resulting in the predicted phase diagram shown in Fig. 9(a). We note that the predictions $\omega_\alpha$ of the 180 CNNs are very consistent: they are constrained in the range $[0, 1]$ and their standard deviation is $\sim 9\%$.

All the points of the SC and CAT phases are correctly classified as non topological and the predicted values of $\tilde\omega$ show a sharp transition between the TRI and TOP phases, meaning that the ensemble was able to learn how to identify the two different phases of the superconductor. On the other hand, there is a less marked transition between the TOP and CDW phases. In Fig. 9(b) we show two horizontal cuts of the phase diagram taken for $\mu = 0.5$ and $\mu = 4$. Interestingly, in the CDW phase for $\mu \gtrsim 1$, the values of $\tilde\omega$ present discontinuities as a function of the potential $V$ (indicated by the dashed vertical lines for $\mu = 4$ in Fig. 9(b)) that correspond to the boundaries of an incommensurate CDW that has been found e.g. in [29]. The appearence of this additional phases is also caught by the PCA (Fig. 3(a, d)) and k-means (Fig. 5(b)) results.

# 5 Conclusions

In this work we have shown how one can use machine learning techniques to predict the topological and non-topological quantum phases of a paradigmatic model, namely the interacting Kitaev chain, by exploiting the knowledge of the easily solvable non-interacting model. More specifically, we have constructed the non-interacting dataset from the analytical solutions of the standard and anomalous correlation functions of the model, while we have used the DMRG algorithm to calculate the correlation functions in the interacting case.

At first, we have probed our data with PCA, k-means clustering and t-SNE. With the former, we have confirmed that the non-interacting data contain enough information to predict the main features of the interacting dataset as well. With k-means, we were able to identify the right number of phases as well as correctly locate the phase transition lines. Also, we have seen that the correlation functions of the four centroids reproduce the pattern of the correlation functions of the different phases. With t-SNE, we were able to see clusters of data corresponding to the phases of the models and, thanks to the silhouette values obtained from k-means, the localization of the phase transition points at the borders of the various clusters.

Then we have used an ensemble of CNN, which was trained on the non-interacting dataset and then tested to calculate and predict the phases of the interacting model. The ensemble does efficiently evaluate the topological class of this model, with only the data points of the topological superconducting phase producing a topological indicator equal to one on average. This means that the ensemble network is able to reconstruct the phase diagram indifferently of the shapes and phases of the input data.

These findings clearly indicate that non-trivial phases of matter that emerge in presence of interactions can be identified by means of unsupervised and supervised techniques applied to data of non-interacting systems. These results offer a number of advantages since the data of non-interacting or solvable quantum many-body systems can be easily generated by means of analytical computation, numerical simulations, or directly measured in state-of-the-art experiments. Furthermore, all the protocols developed in this work could be easily generalized to higher-dimensional systems or finite-temperature regimes.

## Acknowledgements

We thank E. Tignone, F. Dell'Anna, S. Pradhan for useful discussions. This research is funded by Istituto Nazionale di Fisica Nucleare (INFN) through the project "QUANTUM", the International Foundation Big Data and Artificial Intelligence for Human Development (IFAB) through the project "Quantum Computing for Applications" and the QuantERA 2020 Project "QuantHEP". G. M. is partially supported by the Italian PRIN2017 and the Horizon 2020 research and innovation programme under grant agreement No 817482 (Quantum Flagship - PASQuanS). We acknowledge the use of computational resources from the parallel computing cluster of the Open Physics Hub (https://site.unibo.it/openphysicshub/en) at the Physics and Astronomy Department in Bologna.

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
