# Peer review of "Unsupervised and supervised learning of interacting topological phases from single-particle correlation functions"

_SciPost Physics, doi:SciPost Phys. 14, 005 (2023)_

## Round 1 · Referee Report · Anonymous (Referee 1) · 2022-4-12

Strengths

1) Topological phase transitions of interacting models trained on data of non-interacting models are correctly identified. In particular, a topological invariant of an interacting model is predicted by training only on a non-interacting model. This constitutes a very promising result

2) The data used for identification of the topological phases is experimentally accessible, according to the authors. This might open the possibility of application to experiments.

3) The paper makes a nice and exhaustive use of different methods (unsupervised and supervised)

4) The manuscript is very well written and easy to follow

Weaknesses

1) The authors claim, that the topological invariant (winding number) is predicted "with a high degree of accuracy". However, in Fig. 7(b) the winding number for the CDW is estimated lower than 1, but still very visibly nonzero. Also, the transition to the CDW is not very sharp or clear. This lets me hesitate in readily believing in a successful application to more challenging models

2) The benefit of using supervised learning does not become completely clear to me. The transition is also predicted via unsupervised learning (transition to CDW). More accurate results could probably also be obtained by using one of the developed unsupervised techniques (see e.g. A. Dawid et al 2020 New J. Phys. 22 or E. van Nieuwenburg et al Nature Phys 13, 435–439 (2017)) The only additional property that is obtained is the winding number - but as stated above, the applicability beyond this model is a bit questionable to me, if I am not misinterpreting the data. Maybe the transition to the CDW is a very special case?

3) Novelty: Training on exactly solvable models and evaluating on not exactly solvable models is, in contrast to the authors' claim that this goes 'beyond the common scope of machine learning', to my knowledge a commonly used trick, see e.g. Valenti et al, PRR 1(3) (2019). In addition, the method is not applied beyond DMRG results and thus does not yield new physical insights. Furthermore, the authors claim that they use experimentally accessible data in contrast to previous work identifying phase transitions - however e.g. here: Käming, Niklas, et al. "Unsupervised machine learning of topological phase transitions from experimental data." Machine Learning: Science and Technology 2.3 (2021) experimentally accessible data is clearly also used

Report

The authors address the relevant issue of extracting a topological invariant of an interacting model without training on it. In addition, they provide a nice analysis of the used data using unsupervised method. Although the method is not really new, the paper in my opinion could be published in scipost if the authors address the concerns I listed in the section 'Weaknesses' and answer the questions/make the requested changes in the section 'Requested changes'.

Requested changes

  • Is it possible to verify that the learned quantity is really the winding number and not another accompanying trait of this specific model? In particular the nonzero value in the CDW regime and non-sharp transition make me question it. If there is no way of verifying it, an application to a different model could bring evidence and in addition demonstrate, that the manuscript indeed meets the scipost acceptance criteria by 'opening a new pathway in an existing research direction'.

  • I would appreciate a description of the results that is more faithful to the actual results - if I'm not misinterpreting something, the passages 'All the points of the non-topological phases are correctly associated to a zero winding number' and '[...] calculate the winding number [...] with a high degree of accuracy' are just not correct. This request also includes the presentation of 'novelty' (issues addressed in point 3 of weaknesses)

  • An explanation, how the data used for training is experimentally accessible would be a nice addition

  • A minor comment: In Fig. 7, it is a bit confusing that the x-axis of the two subplots is shifted. Would it be possible to align it?

  • validity: good
  • significance: good
  • originality: ok
  • clarity: high
  • formatting: excellent
  • grammar: excellent

Author:  Davide Vodola  on 2022-08-01  [id 2704]

(in reply to Report 1 on 2022-04-12)

We thank the Referee for their careful reading of our manuscript. We appreciate that it was found to be a clear reading with exhaustive analysis and that they found good promises in our work. We address the Referee's comments in the following:

Weaknesses: 1) We thank the Referee for the comment: we agree that using the phrase "high degree of accuracy" could be misleading so we changed the text deleting the sentence. 2) The goal of our work was not to prove the benefits of the supervised method versus the unsupervised ones. Our aim was to understand if also supervised models can be trained and tested on different datasets in order to recognize the phase diagram of the interacting model. 3) We apologize for the possible misunderstanding we might have caused and we thank the Referee for giving the opportunity to clarify this point. We had no intention to state that nobody applied machine learning to experimental data (we do in fact mention explicitly [Torlai2018] as an example of experimental approach). In the Introduction we were referring to the use of synthetic data coming from correlation functions and not from Hamiltonian terms as in [Zhang2018] and [Sun2018]. In addition, we rephrased "This goes beyond the common scope of machine learning" (beginning of Sect. 4) with "This is an approach that has been recently exploited in the context of machine learning applied to systems without analytical solution" adding a reference to Valenti et al, PRR(2019). We also included the reference Käming et al (2021).

Requested changes: 1) We are using the techniques proposed in the manuscript to study the phase diagram of other interesting interacting models, that will be the subject of next papers. In order to convince the Referee that the algorithm can be generalized to different models, we report here the results we obtained by applying it to the SSH model. We generated a training dataset of correlators from the non-interacting Hamiltonian \begin{equation} H_{SSH} = -t_1 \sum_{i=0}^{N/2-1} c^\dagger_{Ai} c_{Bi} -t_2 \sum_{i = 1}^{N/2-1} c^\dagger_{Bi} c_{Ai+1} + h.c. \end{equation} where $N = 100$ sites, $c_{A(B)}$ is the destruction operator of the sublattice $A(B)$-type spinless fermion. We tested an ensemble of networks like the ones in the paper on the training set and then created a test set of correlators adding an interaction term $V = \sum_{i} n_i n_{i+1}$ to the Hamiltonian $H_{SSH}$. We plot the results at two different values of the potential in Fig.1 of this reply (see the attached file). We see that one gets a good agreement with the real values of $\tilde{\omega},$ obtained by looking ad the density of edge state. Let us also notice that, in order to answer the second Referee's comments and to avoid any confusion, we renamed the quantity $\tilde{\omega}$ for the interacting models as {\it topological indicator}, instead of winding number.

2) We explained more carefully which points are classified with high degree of accuracy addressing the problem of the points close to the phase transition between CDW and TOP in the interacting case. Moreover we removed the sentence "high degree of accuracy" in the abstract and the conclusions.

3) We added two recent and more specific references describing two possible experimental protocols [Naldesi et al. arxiv 2205.00981],[Gluza, Eisert PRL 127, 090503] allowing one to measure one-particle correlators.

4) The image was adjusted as requested.

Attachment:

scipost_reply_ref1.pdf

---

## Round 1 · Referee Report · Anonymous (Referee 2) · 2022-4-15

Strengths

1) The paper addresses the classification of topological phases of matter with machine learning. In general, topological phases of matter are a more difficult task for data-driven algorithms as they require the algorithm to learn non-local order parameters. 2) The authors also generalize their methods to the interacting case, which is in itself non-trivial. 3) One key feature is that they choose the input data to be the correlation functions c(k) and f(k). Such quantities can also be computed

Weaknesses

1) The unsupervised learning approach based on PCA is not very convincing 2) The choice of the NN architecture for the supervised approach seems to be quite uncomon.

Report

The authors use different machine learning algorithms to characterize the phase diagram of a topological superconductor. They first start their analysis with unsupervised learning approaches: dimensionality reduction with PCA and clustering with K-means. They then perform a supervised learning approach to infer the winding number when training on the non-interacting model. They then show that such a trained neural network can be then used in the interacting case.

The paper is well written and the results presented here are interesting. From my viewpoint, it, therefore, reaches the standard of publication for Scipost physics. Nevertheless, before recommending the paper for publication, I would like the authors to address several questions/comments that I had when reading the manuscript.

Requested changes

1) The authors should explain more in detail how the quantities $c(k)$ and $f(k)$ are related to the winding number for the non-interacting model. In case there is no direct relation, they should explain why they believe such correlators should give sufficient information to determine the winding number. 2) The authors should comment on the difference between $c(k)$ and $f(k)$ for the non-interacting case, which has been computed for periodic boundary conditions, and for the interacting case which has been computed for open boundary conditions. In particular, I would expect differences coming from the boundary conditions. It would be actually fairer to provide a training set where $c(k)$ and $f(k)$ have been computed for open boundary conditions also for the non-interacting case. 3) Following the indications of the text, one should expect to have a nice clustering In the plane $p_1-p_4$. I, therefore, recommend adding this plot with a coloring that corresponds to the different phases (similar to Fig. 2 of [Phys. Rev. B 94, 195105 (2016)]). I expect to see in such a plot a nice clusterization between trivial and topological and a continuous transition between $\nu=1$ and $\nu=-1$. I recommend doing such a figure for the non-interacting and interacting case. 4) It would be interesting to see whether more powerful dimensionality reduction techniques such as t-SNE or UMAP would allow for a better clustering of the data. 5) In the K-means section, it would actually be instructive to add a similar plot to Fig 4b with the labels of the clusters found by K-means. This could be done for one single run of K-means or with the help of a majority vote. 6) Could the authors also comment on the lines of low values of S in the TRI phases? It seems to me that such lines also appear in Fig. 2b of the PCA analysis.
7) Same comment as 5) for the interacting case. It would be interesting to see the clusterization performed by the algorithm as an additional plot in Fig. 5. 8) The authors should comment on the choice of the NN architecture for the supervised learning scheme. The choice of a 2D convolutional network for 1D-like data is not usual. I would have expected them to use a one-dimensional CNN with two input channels (one for $c(k)$ and one for $f(k)$). 9) Could the authors comment on the reason for the high standard deviation on the training set after training? I would have expected a much smaller standard deviation if the networks were trained properly. Is this high error coming from the points close to the phase transitions? 10) Can the authors confirm that the network is also able to predict a negative winding number in the interacting case?

  • validity: top
  • significance: good
  • originality: good
  • clarity: high
  • formatting: excellent
  • grammar: excellent

Author:  Davide Vodola  on 2022-08-01  [id 2703]

(in reply to Report 2 on 2022-04-15)
Category:
answer to question

We thank the Referee for reading carefully our paper. We appreciate that they recognised the difficulty of this task and appreciated the novelty of using the correlation functions. We address the Referee's comments in the following:

1) The winding number is obtained calculating the integral: \begin{equation} \omega = \int_0^{2 \pi} \frac{h_y(k) \partial h_z(k) - h_z(k) \partial h_y(k)}{E(k)} \end{equation} with $E(k) = \sqrt{h_y^2(k) + h_z^2(k)}$, $h_z(k) = J \cos k + \mu / 2 $ and $h_y(k) = \Delta \sin k$. Since, the correlation functions can be written as:

$$ c(k) = \frac{1}{2} + \frac{\mu/2 + J\cos k}{2 {E(k)}} = \frac{1}{2} + \frac{h_z(k)}{2 {E(k)}}, $$
$$ f(k) = \frac{\Delta \sin k}{2 {E(k)}} = \frac{h_y(k)}{2 {E(k)}}. $$
we can notice that the winding number can be extracted from $c(k)$ and $f(k)$.

2) In general, bulk and global properties of correlators should not depend on the choice of boundary conditions. In order to check this in the model we considered, we show this results in Fig. 1 of this reply (see the attached file) for a system of $L=100$ and different values of the chemical potential.

3) We thank the Referee for pointing out the reference [Wang, PRB 2016] that we incorporated in the reference list of the revised manuscript as Ref. [10]. In Fig. 2 (left panel) of this reply (see the attached file) we show the component $p_4$ ($y$-axis) vs. the component $p_1$ ($x$-axis) for the non interacting data labelled by their winding number. As predicted by the referee, we can indeed see clusterization of the topological trivial vs. the non-trivial phases. However we prefer not to include this plot in the main text of the manuscript as we believe it would provide redundant information. Regarding the interacting model, the clustering of the topological vs non-topological phases is not clearly visible when plotting the component $p_4$ vs. the component $p_1$ (Fig. 2 (right panel) of this reply (see the attached file)).

4) For the sake of simplicity and interpretability we only considered PCA and Kmeans. For completeness in Fig. 3 of this reply we show the clusterization performed by tSNE and we can see that the algorithm recognizes points in different clusters.

5) We thank the Referee for pointing this out. We added the plots as the Referee suggested.

6) We agree with the Referee that these lines are peculiar but unfortunately we do not have a clear understanding of the behaviour of the silhouette at these points. This could probably be due to the change of sign of $\Delta$ which affects the shape of the $f(k)$ correlators. Nonetheless, the values of the silhouettes along those lines are around 0.5 so still very far from 0, which excludes a possibile phase transition.

7) See item 5)

8) The choice of a 2D CNN was motivated by the work of [Zhang et al. PRL(2018)] where the authors train a CNN with a 2D input.

9) We thank the Referee for pointing this out. The sentence is indeed referring to the predictions of the test set. Therefore we moved the sentence to the paragraph Testing.

10) In the model we consider, there is no phase with negative topological indicator. The correlators of the TOP phase of the interacting model, loosely speaking, resemble the correlators of the TOP+1 non-interacting phase as they are computed for $\Delta = +1$. We believe that if the interacting Hamiltonian had $\Delta = -1$ the correlators would resemble the ones of the TOP-1 phase and so the CNN could predict negative winding numbers.

Attachment:

scipost_reply_ref2.pdf

---

## Round 2 · Referee Report · Anonymous (Referee 1) · 2022-9-25

Report

The authors implemented the requested changes mostly as asked. However, the strengths of the manuscript are a bit weakened in my opinion, as it is still not clear whether a topological quantity is predicted (see requested changes 1)). Instead, the authors refer to results of a future paper, not enclosed in the current manuscript. As it is the current manuscript considered for publication, not enclosed results should not matter in the decision for publication.
Nevertheless, as the rest of the manuscript has been adapted as requested and it represents a nice and exhaustive ml-based study of a topological phase transition in the presence of interactions, in my opinion it meets the criteria required for publication. I would thus recommend the paper for publication already in the current state.
If the authors still wish to improve the quality of the manuscript though, including the results they mention in the answer to "requested changes 1)" would be beneficial. This choice, however, I would leave up to the authors.
  • validity: -
  • significance: -
  • originality: -
  • clarity: -
  • formatting: -
  • grammar: -

Author:  Davide Vodola  on 2022-11-18  [id 3044]

(in reply to Report 1 on 2022-09-25)

We kindly thank the Referee for their comments. We greatly appreciate that they saw the validity of our work so that it can be published in the current form. Concerning the point raised about the learned quantity from the model, we understand the importance of such analysis and for this reason we believe it deserves further studies that will be published in a future work.

---

## Round 2 · Referee Report · Anonymous (Referee 2) · 2022-10-13

Report

I thank the authors for the answers to my comments and for the nice revision of the manuscript. Given the nice results for the t-SNE approach in the reply to my questions, I recommend adding a small subsection discussing the t-SNE results for the non-interacting and interacting cases. It would be particularly interesting to see whether the two clusters for phase 0 correspond to TRI- and TRI+.
  • validity: -
  • significance: -
  • originality: -
  • clarity: -
  • formatting: -
  • grammar: -

Author:  Davide Vodola  on 2022-11-18  [id 3043]

(in reply to Report 2 on 2022-10-13)

We thank the Referee for their comments. We are very glad they appreciated the changes we made on the manuscript and we would like to thank them for giving us a chance to improve it. We kindly accept the request of adding the results of the tSNE to the manuscript which now can be found in section IIIC. In particular we can show that the two trivial 0 phases found by tSNE do indeed correspond to the points of the trivial phases TRI+ as TRI- as expected from the Referee.

---

## Round 3 · List of Changes

Added section IIIC on t-distributed stochastic neighbor embedding

---

## Editorial Decision

published